# A novel mouse model for checkpoint inhibitor-induced adverse events

**Kieran Adam[1], Alina Iuga[2], Anna S. Tocheva[1], Adam Mor[1] ***

**1** Columbia Center for Translational Immunology, Columbia University Medical Center, New York, New York, United States of America, **2** Department of Pathology, Columbia University Medical Center, New York, New York, United States of America

* am5121@cumc.columbia.edu

## Abstract

Immune checkpoint inhibitors have demonstrated significant efficacy in the treatment of a variety of cancers, however their therapeutic potential is limited by abstruse immune related adverse events. Currently, no robust animal model exists of checkpoint inhibitor-induced adverse events. Establishing such a model will improve our mechanistic understanding of this process, which in turn will inform design of improved therapies. We developed a mouse model to determine inflammatory toxicities in response to dual checkpoint blockade in the presence of syngeneic tumors. Mice from susceptible genetic backgrounds received intra-peritoneal injections of anti-mouse PD-1 and CTLA-4 antibodies. The mice were monitored for weight loss and histologic evidence of inflammation. Blood was collected for basic meta-bolic panels and titers of anti-nuclear antibodies. In parallel, mice were also treated with prednisolone, which is commonly used to treat immune related adverse events among cancer patients. Among all the genetic backgrounds, *B6/lpr* mice treated with anti-CTLA-4 and anti-PD-1 antibodies developed more substantial hepatitis, pancreatitis, colitis, and pneumonitis characterized by organ infiltration of immune cells. Mice that developed tissue infiltration demonstrated high serum levels of glucose and high titers of anti-nuclear antibodies. Finally, while administration of prednisolone prevented the development of the inflammatory adverse events, it also abrogated the protective anti-tumor effect of the checkout inhibitors. Genetic background and treatment modalities jointly modified the inflammatory adverse events in tumor bearing mice, suggesting a complex mechanism for checkpoint inhibitor-related inflammation. Future studies will assess additional genetic susceptibility factors and will examine possible contributions from the administration of other anti-inflammatory drugs.

## Introduction

Immune checkpoints inhibitors (ICI) increase the survival of patients with multiple malignancies [1]. Immune checkpoints are T cell surface-expressed inhibitory receptors that prevent excessive T cell responses. Tumor cells have developed mechanisms to usurp those inhibitory mechanisms in order to prevent T cell-mediated tumor killing. Initially, the immune system

**Data Availability Statement:** All relevant data are within the manuscript and its Supporting information files.

**Funding:** National Institute of Allergy and Infectious Diseases, AI25640, Dr. Adam Mor National Cancer Institute, CA231277, Dr. Adam Mor National

Institute of Allergy and Infectious Diseases, AI013696, Dr. Adam Mor Cancer Research Institute, Dr. Adam Mor Lisa M. Baker autoimmunity innovation fund, Dr. Adam Mor.

**Competing interests:** The authors have declared that no competing interests exist.

recognizes and eliminates transformed cancerous cells prior to their development into tumors. To escape elimination, tumor cells express inhibitory ligands to prevent T cell recognition [2]. Consequently, the therapeutic blockade of these checkpoints or their ligands with ICI restores anti-tumor immunity. Immune checkpoint blockade for the inhibitory receptors CTLA-4, PD-1, and PD-L1 (the ligand for PD-1) as monotherapy or in combination with other agents have demonstrated improved responses in cancer treatment [3]. The PD-1-PD-L1 interaction directly inhibits anti-tumor T cells responses, promotes peripheral effector T cell exhaustion and enhances Foxp3 expression in Th1 cells [4,5]. These antibodies targeting inhibitory receptors have been approved by the FDA for the treatment of numerous cancers including melanoma, renal cell carcinoma, squamous cell carcinoma, and Hodgkin lymphoma to name a few.

Unfortunately, ICI are associated with significant immune-related Adverse Events (irAEs) as a result of an excessive immune response [6]. Potentially any tissue can be injured as ICI may disrupt self-tolerance to normal tissues. These irAEs range from mild to severe in various tissues, the most common including the skin, liver, lung and gastrointestinal tract [7]. Moderate irAEs requires the temporary discontinuation of ICI and short-term use of corticosteroids with subsequent ICI treatment, therefore limiting their efficacy. Furthermore, severe irAEs can lead to the cessation of therapy altogether, however following established guidelines for managing toxicities allows for rechallenging the tumor with ICI if adverse event grade reverts although treatment should be permanently discontinued for life threatening toxicities [8]. If symptoms do not clearly improve, administration of other immunosuppressive drugs is required such as cyclosporine, mycophenolate mofetil, or infliximab. Additionally, prolonged immune suppression may place the patients at risk for development of infections.

With the advancement in immunotherapy and their enhanced treatment responses, more patients will continually be given these therapies. Unfortunately, there are no simple and clinically relevant animal models for better understanding the pathogenesis of irAEs, assessing the risk of developing severe complications, and testing future interventions. Although most irAEs are low grade [grade 1–2], higher grade events [grade 3–4] can be life threatening and preclude patient's ICI therapy. Thus, there is a strong need to develop pre-clinical mouse models to identify which immunotherapeutic combinations induce the best anti-tumor responses without inducing severe irAEs. To fill this gap, we describe a limited, but feasible and affordable mouse model of irAEs that recapitulates many aspects of irAEs course that occur in humans and would enable us to test the ability and consequences of administration of anti-inflammatory agents in alleviating the symptomatology associated with these adverse events.

## Materials and methods

### General reagents

Dulbecco's modified Eagle's medium (DMEM), Dulbecco's phosphate-buffered saline (DPBS), and fetal bovine serum (FBS) were purchased from Life Technologies. Prednisolone (Sigma) was given orally for 5 days at 1 mg/kg.

### Mouse model

C57BL/6, MRL/MpJ, MRL/lpr, BALB/c, B6/lpr, and SWR/J strains were purchased from Jackson Laboratory (JAX) and used at 6–9 weeks of age. Complete Freund's Adjuvant (CFA) (InvivoGen) was given sub-cutaneously (SC) on days 35 and 56 after initial ICI injection. All animals were maintained in a pathogen-free environment at Columbia University Medical Center. Animal protocols were approved by the Institutional Animal Care and Use Committee of Columbia University Medical Center (AAAW7464). MC38 (Gift from Ben Neel of NYU), a mouse colon adenocarcinoma cell line, was maintained at 37˚C and 5% $CO_2$ in Dulbecco's

modified Eagle medium (DMEM) supplemented with 10% FBS (Corning). C57BL/6 and B6/lpr (JAX) mice were injected at the right flank subcutaneously with $1 \times 10^6$ MC38 cells (B6 background tumors) in 100uL of PBS. Treatment was initiated when tumor volume reached ~100mm$^3$. Tumor volume was calculated with digital caliper using the formula: Tumor volume = (length x width$^2$)/2. In vivo anti-mouse PD-1 (RMP1-14) and anti-CTLA-4 (9D9) from BioXcell were given intra-peritoneally (IP) twice a week at 200 ug and 100 ug respectively, for up to six weeks. Mice were monitored daily for tumor growth, stress or suffering, and body weight was measured twice a week. Humane endpoints to euthanize mice were tumor volumes greater than 2000 mm$^3$, loss of > 20% body weight, and ulceration or severe necrosis of tumor. Mice were euthanized by $CO_2$ asphyxiation followed by cervical dislocation. Less than 10% of the animals were euthanized due to the humane endpoint and no animal died prior to the experimental endpoint.

## Anti-nuclear ELISA

Serum was collected from mice prior to tumor implantation and at endpoint. Anti-nuclear ELISA was performed according to the manufacturer's recommendations (Cusabio Technology) and samples were tested at a 1:200 dilution.

## Immunohistological and image analysis

Tissue was fixed in 10% formalin and processed in the Molecular Pathology Shared Resource of the Herbert Irving Comprehensive Cancer Center at Columbia University for H&E and IHC. IHC anti-CD19 (D4V4B), CD8a (D4W2Z), and CD4 (D7D2Z) antibodies were purchased from Cell Signaling Technology (CST), and F8/40 (BM8) from eBioscience was used at manufacturer's recommended dilutions. Goat anti IgG H+L biotinylated (Vector Laboratories) antibody was used to detect rat IgG anti-mouse PD-1. Slides were scanned using Leica SCN400 and visualized with Aperio ImageScope. Two blinded pathologists assisted with grading of immune infiltration. Two slides were made from each organ. Five random high-power filed images were scored and averaged.

## Hematological studies

Blood was collected prior to treatment and at endpoint. Serum was isolated by centrifugation 10,000 X *G* for 10 minutes and stored in -80˚C until further use. Samples were processed by CUMC Institute of Comparative Medicine (ICM) diagnostic laboratory on Heska Element DC.

## Statistical analysis

Errors bars relate to SEM unless indicated otherwise in figure legend. Statistical testing was performed on GraphPad Prism (Version 8). Statistical significance is indicated as follows: * $p < 0.05$, ** $p < 0.001$, ns not significant.

## Results

### Characterization of immune related toxicities in various genetic strains of mice

First, we wanted to identify the mouse strains that are more susceptible to the development of adverse events. To this end, we tested six strains with different characteristics with PD-1 antibody and CFA (Table 1) and successive experiments with anti-CTLA-4 with anti-PD-1, in the presence or absence of tumor antigen (Table 2). Tissue was processed for H&E and evaluated

**Table 1. Characterization of immune related toxicities in various strains of mice treated with anti-PD-1 antibody and CFA boosters.**

| Mice strain | Immune infiltration | | | |
|---|---|---|---|---|
| | Liver | Colon | Lung | Pancreas |
| C57BL6 | + | - | + | - |
| Balb/c | - | - | - | - |
| SWR | - | + | - | - |
| MRL/mpj | + | - | + | + |

by a pathologist. Treatment with anti-PD-1 & CFA failed to induce immune infiltration in BALB/c mice however, SWR mice showed slight infiltration to the colon (Table 1). In C57BL/6 mice, administration of anti-PD-1 antibodies &CFA resulted in minimal mononuclear cell infiltration to the liver and the lung (Table 1). The presence of MC38 tumors and the addition of anti-CTLA-4 antibodies did not increase the infiltration load in these mice (Table 2). This was in contrast to the autoimmune predisposed MRL/mpj and MRL/lpr mice that demonstrated accelerated infiltration in the context of both anti-PD-1 and anti-CTLA-4 antibodies, emphasizing the contribution of a genetic background to the development of the immune adverse events. To be able to study the contribution of tumor antigens to the immune response in the predisposed mice, and since most murine tumor lines were generated in C57BL/6 mice, we carried out our studies using C57BL/6 mice that were crossed with MRL/lpr mice (B6/lpr). These mice had increased mononuclear cell infiltration in the liver, colon, lung, and pancreas after treatment with both anti-PD-1 and anti-CTLA-4 antibodies and following MC38 tumor challenge (Table 2). IHC on B6/lpr demonstrated major CD4[+] T cells, CD19[+] B cells, and macrophage infiltration.

## Administration of anti-PD-1 and anti-CTLA-4 antibodies results in organ specific leukocytes infiltration in B6/lpr mice

To account for the genetic-drivers that may underlie responses to ICI and to be able to use C57BL6 syngeneic tumor model, we used B6/lpr mice and a continuous administration of anti-PD-1 and anti-CTLA-4 antibodies (Fig 1A). At the end of the experiment, mice were sacrificed, and multiple organs were subjected to histological evaluation. All the mice were injected with MC38 tumors to better model the clinical course of patients with malignancy. While untreated mice revealed no secondary organ infiltration, mice treated with a combination of anti-PD-1 and anti-CTLA-4 antibodies showed hepatitis characterized with heavy infiltration of mononuclear cells to the periportal and pericentral hepatic veins, pancreatitis with perivascular infiltration, colitis with inframammary response in the base of the intestinal villus, and pneumonitis with multifocal peri-bronchial and perivascular inflammation in the lungs

**Table 2. Characterization of immune related toxicities in various strains of mice treated with anti-PD-1 and anti-CTLA-4 antibodies.**

| Mice strain | MC38 | Immune infiltration | | | |
|---|---|---|---|---|---|
| | tumor | Liver | Colon | Lung | Pancreas |
| C57BL6 | + | + | - | + | - |
| MRL/mpj | - | + | - | + | + |
| MRL/lpr | - | + | + | + | + |
| B6/lpr | + | + | + | + | + |

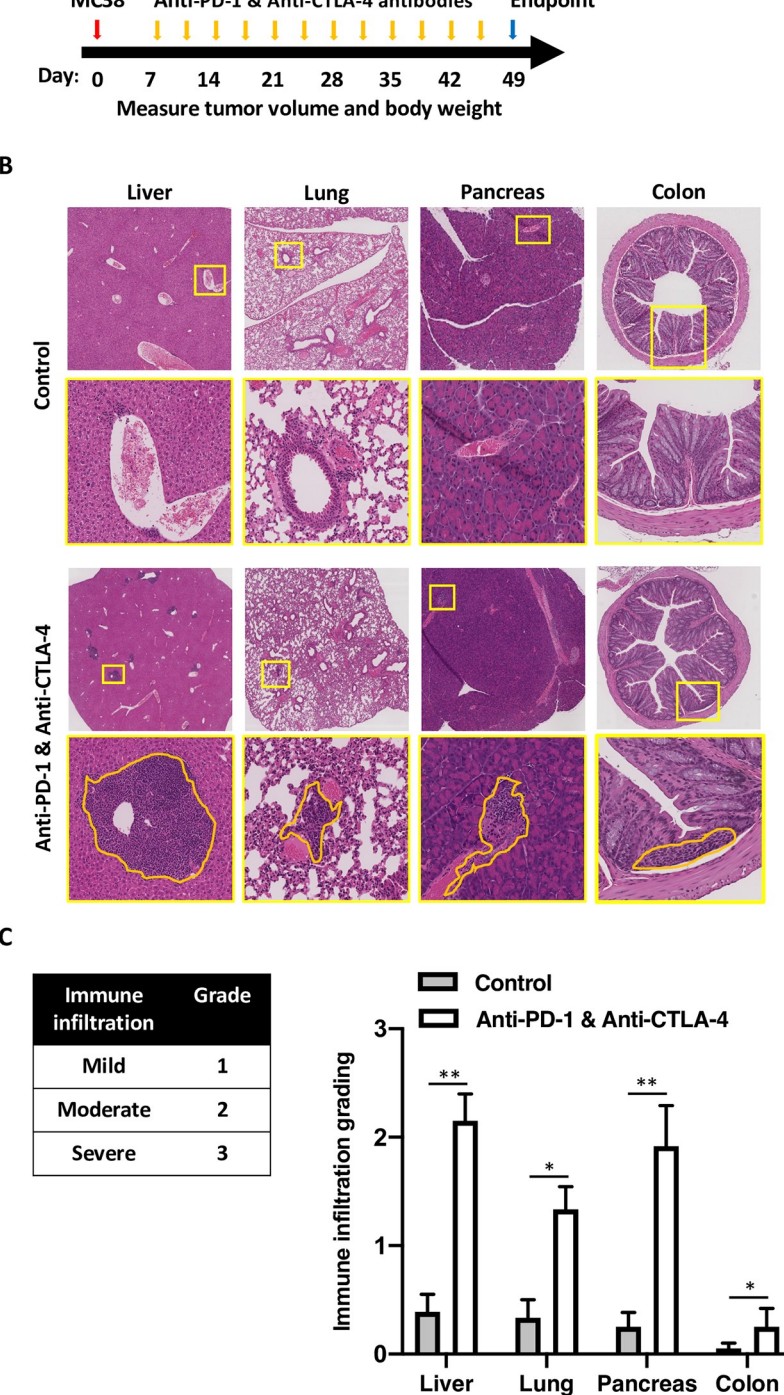

**Fig 1. Administration of anti-PD-1 and anti-CTLA-4 antibodies results is organ specific leukocytes infiltration in B6/lpr.** (A) a diagram showing the syngeneic tumor model protocol and drug administration used in this figure. MC38 tumors were inoculated in day zero, and anti-PD-1 and anti-CTLA-4 antibodies were given twice a week. The experiments were terminated after 49 days. (B) Liver, lung, pancreas, and colon were collected from all the mice at the end of the experiments and stained with H&E. Representative images are shown. Yellow lines represent the areas that were magnified, orange lines represent the mononuclear cell infiltrates. Treatment conditions are as indicated; control mice were treated with isotype control antibody. (C) A table showing the grading system used to quantify the amount of the mononuclear cell infiltrated (left). Quantification of the immune infiltrates per each organ, averaging five high power fields per mouse, 5 mice per group, average ± SEM, * p < 0.05, ns not significant.

(Fig 1B). Grading of these infiltrates by two independent blinded scientists showed significant infiltration to the liver, lungs, pancreas, and colon (Fig 1C). In all mice, the skin, hearts, and joints were free of inflammation.

## Inverse correlation between tumor size and secondary organ infiltration in mice treated with anti-PD-1 and anti-CTLA-4 antibodies

One of the most relevant clinical question in treating patient with ICI is whether the onset or severity of irAEs are related to the protective anti-tumor response rate of the same antibodies. In MC38-B6/lpr mice treated with anti-PD-1 and anti-CTLA-4 antibodies the discrimination between responders and non-responders was determined by observing the segregation of the tumor growth curves between both populations (Fig 2A). Similarly to cancer patients treated with ICI, only one third of the mice responded to the treatment. There was no statistically significant difference in the body weight of respondent vs. non-respondent mice at the end of the experiment (Fig 2B). However, histological analysis clearly demonstrated increased immune infiltration in the liver, lung, pancreas, and colon of those mice whose tumors responded favorably to PD-1 and CTLA-4 blockade (Fig 2C and 2D). Furthermore, while all mice treated with ICI combination had low levels of anti-nuclear antibodies (ANA) [9], those mice that responded to the treatment had even higher ANA levels, suggesting that these infiltrates might have represented the onset of autoimmune inflammation (Fig 2E). Next, to correlate the amount of the infiltration in the pancreas with functional parameters and organ function, serum glucose levels were measured to show high levels in the responders compared to the non-responders (Fig 2F). Interesting all mice had normal liver function (S1 Table). Finally, significant inverse correlation was found between tumor volume and grade of secondary organ infiltration (Fig 2G). Altogether, this data suggests that similarly to cancer patients that receive ICI, mice that develop irAEs respond better to ICI inhibition, intuitively explained by the ability of ICI to re-energize effector T cells [10] with different antigen-specificities.

## Organ-specific composition of cellular infiltration secondary to PD-1 and CTLA-4 blockade

Since T cells are the predominant immune cell population expressing PD-1 and CTLA-4, we performed immunohistochemistry staining of harvested tissues to further characterize the infiltrating T cell populations in mice that responded to the ICI combination. The majority of immune cells in the liver, lung and pancreas were CD4$^+$ T cells with a smaller proportion of CD8$^+$ T cells (Fig 3). Significant number of CD19$^+$ B cells and F8/40 macrophages were also recorded in the liver and in the colon. Staining of draining lymph nodes collected from these mice showed mixed pollution of immune cells. To mechanistically correlate these findings with the anti-PD-1 and anti-CTLA-4 antibodies that were given to the mice, we stained the tissues with anti-rat IgG H+L antibody (Fig 3; lower panel). Interestingly, it was mainly the CD4$^+$, and not the CD8$^+$ T cells, in the liver that bound the therapeutic antibodies. The majority of the administered antibodies were absent from the lung infiltrates and were completely missing in the pancreas and in the lymph nodes. Altogether, this data suggests that the mechanism of irAEs may be distinct in different secondary organs.

## Steroid treatment reduces both T cell infiltration and anti-tumor response

Steroids are the drug of choice to treat patients with irAEs [8]. Clinical data strongly supports the anti-inflammatory effect of steroid on improving both signs and symptoms of irAEs, although it is not clear whether it interferes with the protective pro-inflammatory and anti-

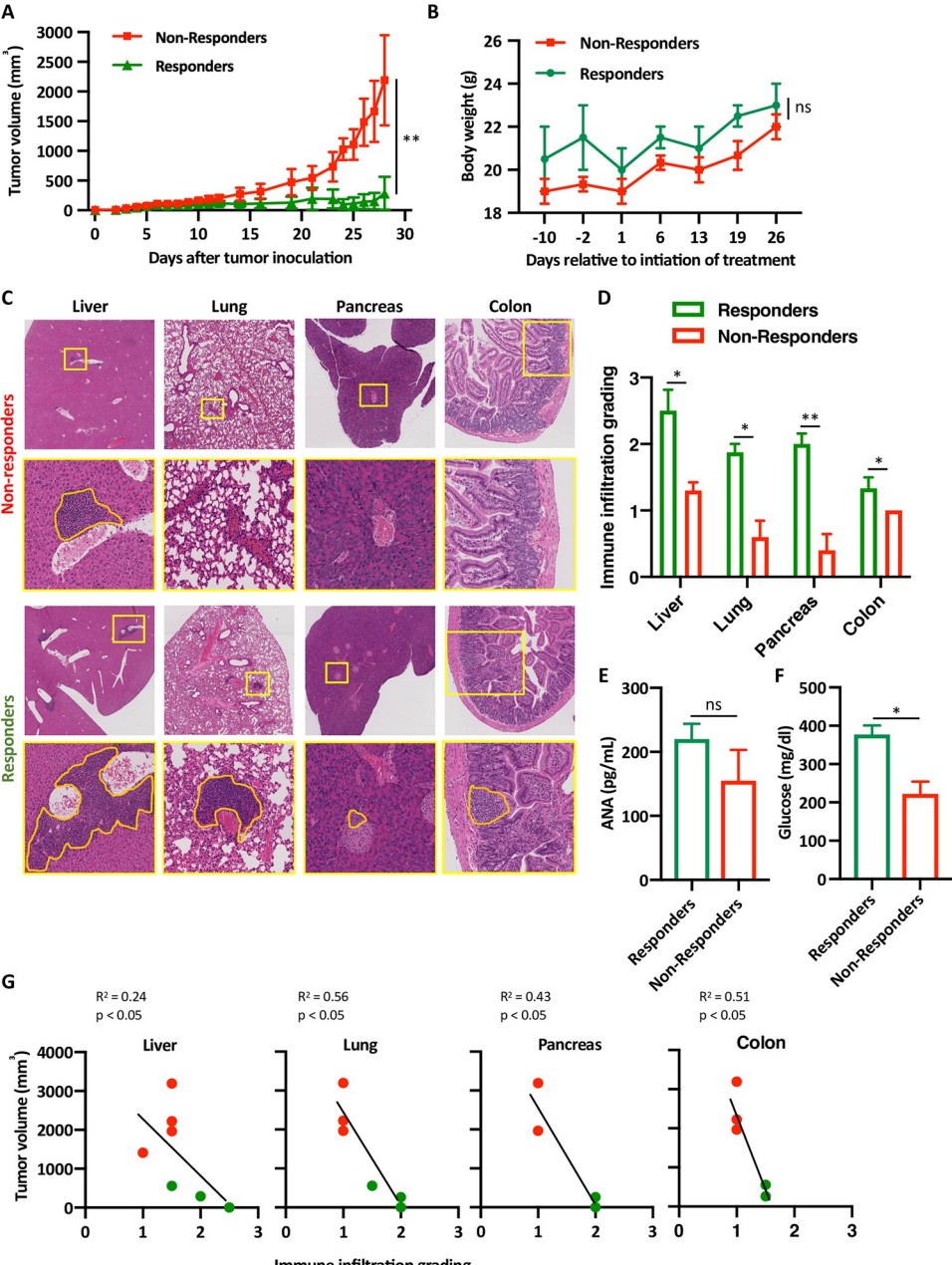

**Fig 2. Inverse correlation between tumor size and organ infiltration in mice treated with anti-PD-1 and anti-CTLA-4 antibodies.** (A) B6/lpr mice were injected with MC38 tumors and tumor size were measured daily. *t*-test was used to compare the average tumors size among mice that responded to the treatment with anti-PD-1 and anti-CTLA-4 antibodies (green) and mice that failed to respond to the same treatment (red). Five mice were in each group, n = 3, ** p < 0.001. (B) Mice that either responded to the treatment (n = 5), or not (n = 15) were weighted and average weights are shown ± SEM, ns not significant. (C) Representative H&E staining images of the liver, lung and pancreas of mice from the non-responders group and the responder group. Yellow boxes are area of magnification; orange lines represent area of immune infiltrates. (D) Quantification of the same immune infiltrates. For each organ, 5 random high-power filed were scored from each mouse; 3 mice per group (altogether 15 high-power fields per condition). Average scoring is shown ± SEM, * p < 0.05. (E) Anti-nuclear antibodies (ANA) titers were measured from mice sera at the end of the experiments from either responders or non-responders using ELISA, n = 5, average ± SEM. (F) Glucose serum levels were measured from responders vs. non-responders at the end of the experiments, n = 4, average ± SEM, * p < 0.05. (G) Pearson correlation coefficients between tumor volume and immune infiltrates grading in the liver, lung, colon and pancreas are shown, each red dot represent mouse that failed to response, green dots represent mice that responded to the treatment with anti-PD-1 and anti-CTLA-4 antibodies.

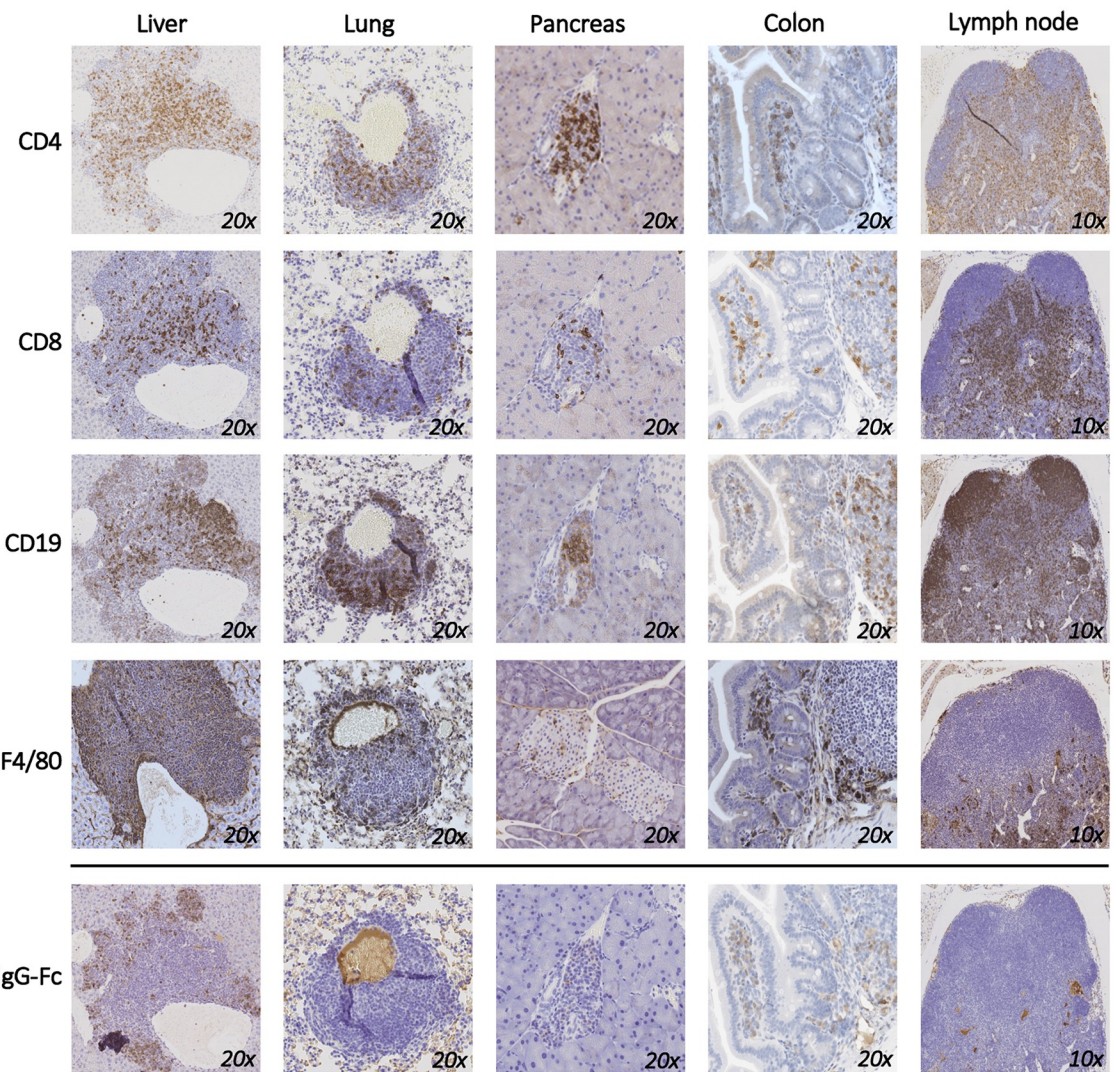

**Fig 3. Organ-specific composition of cellular infiltration secondary to PD-1 and CTLA-4 blockade.** Liver, lung, pancreas, colon, and lymph nodes were collected from mice that responded to the treatment with anti-PD-1 and anti-CTLA-4 and immunohistochemical staining for CD4, CD8, F4/80, and CD19 are shown. Anti-Rabbit Fc staining was used to show the distribution of the therapeutic monoclonal anti-PD-1 and anti-CTLA-4 antibodies used to treat the mice. A representative image is shown out of 3 mice that were used for this experiment.

tumoral immune response of the ICI. Consequently, we modified our protocol and treated the mice with prednisolone for 5 days (Fig 4A). Similar to our previous observations, mice treated with anti-PD-1 and anti-CTLA-4 antibodies segregated into responders and non-responders (Fig 4B). Interestingly, the growth of the tumors among the mice that were treated with ICI and prednisolone, was higher in comparison to the responder group and lower than the non-responders (Fig 4B). There was segregation within the group treated with ICI and predniso-lone, however, the tumor growth curve was a median between the non-responders and responders. This data suggest that the administration of steroids may interfere with the favor-able anti-tumor efficacy of ICI. Remarkably, histological analysis of end organs in these mice reveled significant reduction in the immune infiltration (Fig 4C), as objectively quantified (Fig 4D).

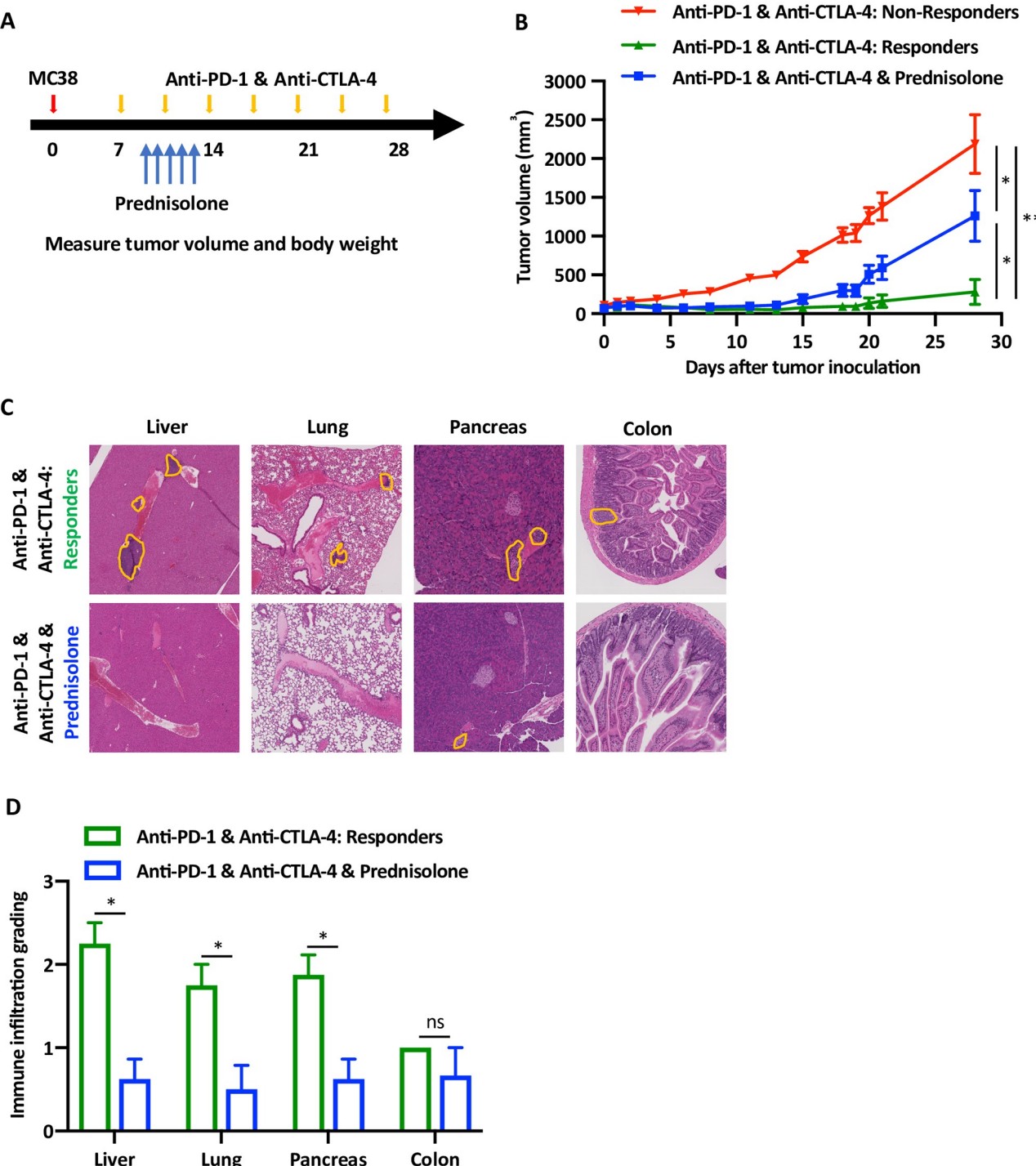

**Fig 4. Steroid treatment reduces both T cell infiltration and anti-tumor response.** (A) Experiential design of the syngeneic MC38 tumor model and the timing of drug administration, including anti-PD-1 and anti-CTLA-4 antibodies and the prednisolone (1 mg/kg). (B) Tumor growth curves of the mice divided per treatment groups, five mice in each group, n = 3. Red and green lines represent average ± SEM tumor volumes of the mice that either responded or not. The blue line represents the average ± SEM tumor volumes of the mice that were treated with prednisolone, * $p < 0.05$, ** $p < 0.001$. (C) Representative H&E staining of liver, lung, pancreas and colon of the mice according to the treatment groups, as indicates. Orange lines show areas of immune infiltration. (D) Quantification of the data shown in the previous experiment. Grading of 5 high power fields from 3 mice were averaged ± SEM. * $p < 0.05$, ns not significant.

## Discussion

The past decade has seen an exponential growth in the development and use of immunotherapy for the treatment of cancer especially immune checkpoint receptor blockade. Unfortunately, effectiveness is limited not just based on the response to the treatment, but also by the occurrence of adverse events [9,10]. There are very few known markers to predict for irAEs and their severity, limiting their therapeutic efficacy [11,12]. In a recent innovator study, the levels of anti-BP180 (hemidesmosomal proteins) IgG correlated to the development of skin related irAEs in non-small cell lung cancer patients [13]. Parallel to that, there is a need to develop a mouse model of irAEs to screen for anti-tumor efficacy as well as severity of irAEs. In this report, we describe a novel mouse model we developed.

We treated *B6/lpr* mice [14] with anti-PD-1 and anti-CTLA-4 antibodies in the setting of established MC38 colon adenocarcinoma. As with patients, this treatment protocol resulted in inhibition of tumor growth in 30% of the cases [15]. Notably, the same 30% of the mice that responded favorably to the treatment with the antibodies also demonstrated secondary inflammatory responses in the liver, pancreas, colon, and lungs. This suggests that reduction of tumor size is a clinical marker for the development of irAEs. To further support that, early reports from lung cancer and melanoma patients demonstrated a similar correlation between the anti-tumor effect of ICI and the severity of the irAEs [16–18]. A proposed mechanism of irAEs is that ICI reduce the self-tolerance of central memory or tissue resident T cells leading to the infiltration of tissues [19]. Human data has also informed us that ICI activates more than one population of T cells. Whether this process is antigen specific or not is unclear. Recent work proposed that the inflamed secondary organs share antigenic epitopes with the transformed cancer cells, suggesting that ICI lowers the threshold of antigen-specific T cell activation [20]. In their report, patients with lung cancer had more pneumonitis. Similarly, it has been reported that patients with melanoma developed more vitiligo, an autoimmune inflammatory skin condition [21,22]. A shared epitope model is not supported by our data as a founder event since the apparent irAEs also occurred in mice that did not endure tumors.

While it might be a limitation of our model, the genetic background of the mice is also a contributing factor. As we demonstrated through this work, the *lpr* mice, which are prone to develop autoimmune responses, developed secondary organ-inflammation [23,24]. Likewise, it has been shown that patients with established autoimmune diseases had more flares while on ICI [25]. It is not entirely clear the mechanism and whether pre-clinical autoimmunity is a risk factor for the development of irAEs in cancer patients, and active work in our own laboratory is focused on deciphering that.

We used IHC to discover which cells bound the ICI that were therapeutically given to the mice. We found that T cell in the liver bound to the anti-PD-1 and anti-CTLA-4 antibodies, but not the cells in lung or the pancreas at the end of the experiment. In addition, most of these cells were CD4$^+$ and not CD8$^+$. This suggests that the mechanism of irAEs in the liver and the other organs is likely to be different, highlighting the contribution of the local microenvironment. Immune infiltration of the tumor was not assessed in this experiment. It would be interesting to see if the tumor had similar subset infiltration as the tissue, furthermore single cell T cell receptor (TCR) sequencing of the tumor infiltrating cells and tissue could characterize the TCR repertoire of these cells. It is important to mention that higher grade irAEs were observed when anti-PD-1 and anti-CTLA-4 were combined, similarly to patient responses to combination ICI therapy. Altogether, it is likely that the development of irAEs is a multifactorial process leading to patient-specific organ involvement [26].

Our model provides, limited, but valuable insights into the anti-tumor effects and irAEs severity associated with novel combination immunotherapies. This may aid clinicians and

pharmaceutical establishments in clinical trial design. It is also relatively simple and can be adopted by many investigators. As a proof-of-concept for drug testing platform, we showed that treatment with steroids prevented the onset of the irAEs, but also interfered with the beneficial anti-tumoral effect of the ICI. Better controlled experiments are needed to inform clinicians about the safest anti-inflammatory approaches to treat these patients.

Interestingly, a recent study demonstrated that TNFα inhibitors concomitantly with combined anti-PD-1 and anti-CTLA-4 antibodies improved immune related colitis in addition to enhanced anti-tumor efficacy [27,28]. This was shown using a model in which immune deficient mice were adoptively transferred with human PBMC, causing graft-versus-host like disease that was further worsened by anti-PD-1 and anti-CTLA-4 antibodies treatment. When human colon cancer cells were xenografted into these mice, the prophylactic blockade of TNFα improved the colitis and the tumors were retained. In contrast, our results failed to show that steroids dissociated the efficacy from the toxicity of the ICI.

Other more complicated and costly models have been published. Treg depletion characterize both anti-tumor responses and severity of irAEs. A recent elegant model took advantage of tumor-bearing Foxp3-DTR mice to deplete these cells in anti-PD-1 and anti-CTLA-4 treated mice [29]. Treg depletion lowered the immune tolerance threshold and allowed irAEs to be induced more easily following treatment with ICI. In this model, the irAEs appeared because of an infiltration of effector T cells in the tissues, however, TNFα blockade decreased the irAEs severity without impacting tumor growth rate. One of the strengths of this model is its ability to deplete Treg by a single injection of DT, making the mice more sensitive to irAEs development from clinically relevant therapies. However, the Treg depletion was not demonstrated in all types of irAEs and Foxp3-DTR mice are not accessible to most investigators.

In a similar study to ours, Korman et al., described a pre-clinical system using anti-PD-1 and anti-CTLA-4 antibodies in MC38 and CT26 syngeneic tumor models [30,31]. Significant anti-tumor activity was demonstrated using variable doses of anti-CTLA-4 and anti-PD-1 antibodies. Only gastrointestinal irAEs were observed with combination treatment in cynomolgus macaque study, supporting our data of limited onset of irAEs using single agent.

Despite the knowledge that type I diabetes (TID) is a rare irAEs in human, several groups used the NOD mice to model secondary organ immune infiltration [32,33]. In these studies, inhibition of PD-1 signaling in NOD mice accelerated the onset of TID. The NOD ldd cogneic mice strains developed TID although these mice had resistant ldd genes (ldd5, ldd3/10/18, and ldd9) however only the ldd3/5 strain was protected from disease onset with PD-L1 blockade. This data indicates the correlation of ldd loci with PD-1 signaling and that PD-L1 blockade impairs the ldd genetic resistance in NOD model of TID.

Finally, a more recent study reported an irAEs model using mice harboring the humanized *Ctla4* gene [34]. Humanized mice treated with Ipilimumab (anti-CTLA-4) and anti-PD-1 antibodies developed severe irAEs. Flow cytometry data of T cell frequencies indicated significant activation Tem (CD44$^{hi}$CD62L$^{lo}$) and reduced ratios of Treg to Teff in auto-reactive CD4 T cells, however, humanized clones of CTLA-4 antibody had similar anti-tumor efficacy and improved safety. This paper demonstrated that the development of irAEs is independent from the blocking of CTLA-4-B7 interaction and the expansion of the T cells.

## Conclusions

The revolution in innovative cancer immunotherapies has resulted in astonishing clinical successes in the treatment of multiple malignancies [35–37]. ICI that target inhibitory receptors have become standard of care for a variety of cancers. However, with the growing use of ICI, alone or in combination with chemotherapy, targeted therapies, or other immune modulators,

a significant increase in irAEs has occurred. Growing evidence indicates that many irAEs are a consequence of a breakdown in tolerance, but many questions related to the pathogenesis and clinical care remain unsettled [38]. Our work describes a mouse model of irAEs that will assist in the validation and efficacy testing of novel anti-cancer immunotherapies and uncover the role of the immune response in mediating the associated toxicities, which will lead to better overall insight into the mechanism of irAEs development and treatment.

## Supporting information

**S1 Table. Basic metabolic panel of mice with multi organ immune infiltration.**
(PDF)

**S1 Data.**
(PDF)

## Author Contributions

**Conceptualization:** Anna S. Tocheva, Adam Mor.

**Data curation:** Kieran Adam.

**Formal analysis:** Kieran Adam, Alina Iuga, Adam Mor.

**Funding acquisition:** Adam Mor.

**Investigation:** Kieran Adam.

**Methodology:** Kieran Adam, Adam Mor.

**Supervision:** Adam Mor.

**Writing – original draft:** Kieran Adam, Adam Mor.

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
