## [Decision Letter · Decision Letter 0]

2 Oct 2020

PONE-D-20-17843

A novel mouse model for checkpoint inhibitor-induced adverse events

PLOS ONE

Dear Dr. Mor,

Thank you for submitting your manuscript to PLOS ONE. After careful consideration, we feel that it has merit but does not fully meet PLOS ONE’s publication criteria as it currently stands. Therefore, we invite you to submit a revised version of the manuscript that addresses the points raised during the review process.

We consider that the manuscript lacks precision in various aspects, and the data presented need to be  completed and strengthened. We invite you to submit a revised version of the manuscript that addresses all the points raised by the reviewer (see below the Reviewers' comments).

Please submit your revised manuscript within 2 months. If you will need more time than this to complete your revisions, please reply to this message or contact the journal office at plosone@plos.org. Please include the following items when submitting your revised manuscript:

We look forward to receiving your revised manuscript.

Kind regards,

Prof. Pierre Bobé

Academic Editor

PLOS ONE

Journal Requirements:

'This work was supported by grants from the NIH (AI25640, CA231277, CA013696) and the Cancer Research Institute.'

'NIH AI25640 to A.M.

NIH CA231277 to A.M.'

3. Thank you for stating the following in your Competing Interests section: 'No'

4. Please include your tables as part of your main manuscript and remove the individual files.

Please note that supplementary tables should be uploaded as separate "supporting information" files.

Reviewers' comments:

Reviewer's Responses to Questions

**Comments to the Author**

1. Is the manuscript technically sound, and do the data support the conclusions?

Reviewer #1: Partly

2. Has the statistical analysis been performed appropriately and rigorously? 

Reviewer #1: Yes

3. Have the authors made all data underlying the findings in their manuscript fully available?

Reviewer #1: Yes

4. Is the manuscript presented in an intelligible fashion and written in standard English?

Reviewer #1: Yes

5. Review Comments to the Author

Reviewer #1: The paper by Adam et al, entitled “A novel mouse model for checkpoint inhibitor-induced adverse events” aims at providing a mouse model for investigating immune related adverse events occurring during treatment with immune checkpoint blockade (ICB). This paper addresses an unmet need in further advancing research in this emerging field as – outlined by the authors – this is currently mostly limited to retrospective clinical investigations. After assessing several mouse strains for suitability, the authors chose to use the B6/lpr strain. The strain presents with autoimmunity and is implanted with with MC38, a highly immunogenic adenocarcinoma of the colon. Following treatment with anti-PD1 and anti-CTLA4 antibodies, infiltration of mononuclear cells to organs such as lung liver and pancreas are observed. This was considered a surrogate for irAE. Whereas the addressed question is of highest importance, the presented data does unfortunately not fully support the message of the paper. The investigation was undertaken in a mouse strain with known autoimmunity, likely limiting any translational value to patients with known autoimmune disease prior to initiation of ICB. Furthermore, levels of infiltration by mononuclear cells in certain organs is not a sufficiently funded surrogate of irAE. Notably, organs frequently affected by irAE in humans such as the skin, heart and joints were free of infiltration in this model, or not assessed (thyroid and other endocrine organs). The paper in the current form doesn’t warrant publication.

Other comments:

56- Melanoma should be included here as this has been the first approved indication.

59- T-cell responses are likely to play a key role in irAE, however it remains unclear whether this is the only driving cell type.

60- Fortunately, with appropriate management – and depending on grade of AE -, treatment needs to be permanently held only in a minority of patients (as described by the authors further on).

66- This should be kept broader as after failure to respond to (high dose) corticosteroids, other immune modulating agents such as MMF, cyclosporine and TNF inhibitors can be employed. TNF inhibitors are not commonly used in ir hepatitis as they are potentially hepatotoxic.

87 onward C57BL/6 and B6/lpr mice were injected with MC38, followed by CPI. Were the other strains injected as well?

90 Please clarify the timepoint of CFA injection (e.g. day x after first CPI treatment/tumor injection etc.)

143 Please specify if this is the case in all strains/with all treatments.

147 Based on IHC, please specify the types of infiltrating mononuclear cells.

157 Please state if CFA was injected in both ‘treated’ and ‘untreated’ mice.

163 This is an interesting finding that should be further discussed as it includes sites of most frequent irAE in humans and might – or might not – be due to effects mediated by other cells and mechanisms than T cells.

175 Specify ‘immune infiltration’ e.g. of tumors or off-target organs.

183 This relates to figure 2F, not 2D

212 Please specify if there is a segregation within the group concurrently treated with prednisolone!

219 Less strong terminology than ‘no’ would be adequate as there are some limited baseline predictive markers such as levels of BP180 antibodies.

220 This sentence is not easily understood.

232 This sentence is hard to understand.

252 It should be stated whether this is found in treated tumors.

312 In lieu of references 32/33, reports of pivotal trials or robust review articles might be preferable.

Table 2 Immune infiltrate to the colon is reported here as opposed to the absence of infiltrate reported in the results section and in Figure 1.

Figure 4

The title is misleading, as prednisolone is reducing both tumor control and T cell infiltration in other organs in this experiment.

6. PLOS authors have the option to publish the peer review history of their article (what does this mean?). If published, this will include your full peer review and any attached files.

Reviewer #1: No

---

## [Author Response · Author response to Decision Letter 0]

15 Dec 2020

The paper by Adam et al, entitled “A novel mouse model for checkpoint inhibitor-induced adverse events” aims at providing a mouse model for investigating immune related adverse events occurring during treatment with immune checkpoint blockade (ICB). This paper addresses an unmet need in further advancing research in this emerging field as – outlined by the authors – this is currently mostly limited to retrospective clinical investigations. After assessing several mouse strains for suitability, the authors chose to use the B6/lpr strain. The strain presents with autoimmunity and is implanted with with MC38, a highly immunogenic adenocarcinoma of the colon. Following treatment with anti-PD1 and anti-CTLA4 antibodies, infiltration of mononuclear cells to organs such as lung liver and pancreas are observed. This was considered a surrogate for irAE. Whereas the addressed question is of highest importance, the presented data does unfortunately not fully support the message of the paper. 

We would like to thank the reviewer for his effort to make this work stronger. We agree with the reviewer that the paper has limitations, but we were very careful not to draw specific conclusions that were not supported by data. Conceptually, this is an observational paper describing the use of dual checkpoint inhibition in mice that develop immune infiltration in several organs. These observations are reproducible and accordingly constitute the main message of the paper. In the revised version, and as suggested by the reviewer, we provide additional histology (H&E) and IHC data to further support the message and also considerably modified the discussion. Furthermore, we added new references to emphasis not just the strengths but also the limitations of this model. 

The investigation was undertaken in a mouse strain with known autoimmunity, likely limiting any translational value to patients with known autoimmune disease prior to initiation of ICB. 

We agree with the reviewer that this is a limitation of the model. We updated the text carefully to further emphasis this limitation with the readers. However, the model does have a significant translational value based on the following arguments: 

1) Mice that are homozygous for the lymphoproliferation (lpr) mutation can show autoimmunity, but the onset and severity is strain-dependent (Nose et al. Rev Immunogenet 2000). While MRL/lpr develop systemic autoimmunity at 6 months of age, B6/lpr, the strain that was used in this work, develop autoimmunity at 15 months of age. Our mice did not have autoimmunity during the time frame of the study. We started the experiment with 6 weeks old mice and terminated it before the mice reached 4 months of age. The control group did not develop clinical or subclinical (serological and histological) autoimmunity. 

2) Most of the animal models of inflammatory and autoimmune diseases are done in mice with specific autoimmune genetic background, even if this information is not apparent (Lee et al. Clin Rev Allergy Immunol 2013). CIA model is taking advantage of DBA mice, type I diabetes is using NOD mice, and some of the lupus model are using NZB/W mice. Our model in no exception. We used B6/lpr and not MRL/lpr mice to minimize the contribution of the genetic background to the phenotype. Unfortunately, wildtype mice of different genetic backgrounds, failed to develop any type of toxicities (Table 1), experimentally suggesting that the genetic background is actually required for irAEs. PD-1 knockout mice develop autoimmunity at 6 months of age, but it is not practical to recapitulate that by treating wildtype mice with anti-PD-1 antibodies for six months after inoculation of tumors that limit mice survival and establish that as a model of a diseases. 

3) Genetic susceptibility is an integral element in the development of clinical autoimmunity in human and mice. Genetic background is specifically important in the context of PD-1 (Pedoeem et al. Clin Immunol 2014). PD-1 knock out mice develop late onset autoimmunity grounded on their genetic background, that will determine the involved organs. For example, in the BALB/C PD-1 knockout mice, a fatal dilated cardiomyopathy has been reported. Interestingly, no such involvement is available on the more commonly utilized PD-1 knockout mice on the C57BL/6 background, that in return develop enlarged spleen, hepatitis and diabetes. It is important to mention that PD-1 knockout mice develop signs of autoimmunity at very late age. 

4) To address the reviewer’s concern, and similar to the patients that we see in the Rheumatology clinic, it is likely that many, if not the majority of the patients that develop immune related toxicities with ICB carry autoimmune susceptible genes. This premise is based on data that was recently reported in a review manuscript that summarized 129 papers that demonstrated association between susceptible inflammatory genes and the occurrence of irAEs among cancer patients (Hoefsmit et al. ESMO 2019; Weinmann et al. Rheumatology 2019). Many of the reports showed risk loci that are shared between primary autoimmunity, and that are known to affect immune functions, such as antigen presentation, cytokine signaling, and regulation of T cell functions. Larger prospective studies will need to support that, but it is reasonable to hypothesis that irAEs patients have genetic susceptibility, but no clinical autoimmune disease prior to ICB. Based on a parallel NCI funded project in our lab, I can confirm that patients with grade 3-4 irAEs have a very unique transcriptome signature prior to the onset of the treatment, emphasizing, again the possible genetic susceptibility of the involved patients. 

Furthermore, levels of infiltration by mononuclear cells in certain organs is not a sufficiently funded surrogate of irAE. 

We agree with the reviewer that infiltration to the liver, pancreas and lungs are not the best surrogate for irAEs. However, and compared to other irAEs animal models, it does have some advantages. The main advantage is its objectivity. It can also provide information about the specific subset of inflammatory cells infiltrating the organs. For this revision, and as suggested by the reviewer, we stained the tissues also for F4/80 and report heavy macrophages infiltration in the context of immune checkpoint blockade, suggesting that irAEs are not limited to the T and B cell compartments. Infiltration of tissue by mononuclear cells is indicative of abnormal immune response, therefore these findings might have clinical implications. The immune infiltrates to the pancreas resulted also in elevated blood glucose levels. It is very likely that if we had prolonged the model by 4 to 6 additional weeks, mice would have been developed inflammatory symptoms, however the limiting factor here was the fact that mice were inoculated with tumors that largely resulted in early death, preventing us to rely on clinical phenotype. We also observed some differences in the weight of the mice and with ANA titers. 

Notably, organs frequently affected by irAE in humans such as the skin, heart and joints were free of infiltration in this model, or not assessed (thyroid and other endocrine organs). 

The reviewer is correct, and the most prevalent irAEs to monotherapy with anti-PD-1 antibodies involve gastrointestinal, skin, endocrine, and liver toxicities, while myositis, arthritis, neuropathies and nephritis are far less frequently reported. It must be emphasized that the frequency of the involved organs in patients treated with combination of anti-PD-1 and anti-CTLA-4 (as our mice) are not the same. Hepatitis is actually common among cancer patients treated with both anti-PD-1 and CTLA-4 and has been reported in up to 17% of all patients (DeMartin et al. JHEP 2020; Reddy et al. Clin Transl Gastroenterol 2018; https://www.uptodate.com). No inflammation of the skin or joints were observed by clinical evaluation during the time of the study. Unfortunately, the thyroid and pituitary were not assessed in this study, due to the low frequency of involvement in lpr mice overall. In addition to the liver, pancreas, and lungs, the colon was also inflamed in our model, and this information has been recalculated and is now included in the revised manuscript (Table 1). 

The paper in the current form doesn’t warrant publication.

We hope that our arguments, the additional histological data provided, and the revised text will convince the reviewer that this paper worth publication in PLoS one. We agree that the model has limitations, but its simplicity, reproducibly, and its low cost will contribute to its popularly among the scientific community. Limitations are common to many experimental models, and as long as the investigators are aware of that and avoid drawing overstated conclusions, its usage is likely justified. In the revised version, we updated the text in order to highlight not just the strengths, but mainly limitations of this model. 

Other comments:

56- Melanoma should be included here as this has been the first approved indication.

Thank you. The reviewer is correct and we added that to the text.

59- T-cell responses are likely to play a key role in irAE, however it remains unclear whether this is the only driving cell type.

We completely agree, and similar to other type of inflammatory responses, it is likely that other cells also play a role. This is now mentioned in the text and supported by the new set of data where we stained the tissues for macrophages (F4/80) and B cells (CD19).

60- Fortunately, with appropriate management – and depending on grade of AE -, treatment needs to be permanently held only in a minority of patients (as described by the authors further on).

We agree with this comment, and the text was modified accordingly. 

66- This should be kept broader as after failure to respond to (high dose) corticosteroids, other immune modulating agents such as MMF, cyclosporine and TNF inhibitors can be employed. TNF inhibitors are not commonly used in ir hepatitis as they are potentially hepatotoxic.

The reviewer is correct that in some settings TNF inhibitors are not the drug of choice. Based on the National Comprehensive Cancer Network, steroids are the gold standard treatment followed by either anti-TNF or anti-CD20 depending on the tissues involved (https://www.nccn.org/patients/guidelines/content/PDF/immunotherapy-se-ici-patient.pdf). In our irAEs clinic at Columbia University Medical Center, we use anti-TNF for many adverse events, but in the case of hepatitis, Imuran or MMF are the drugs of choice. The text was updated accordingly, thank you.

87 onward C57BL/6 and B6/lpr mice were injected with MC38, followed by CPI. Were the other strains injected as well?

No other strains were injected as this is a syngeneic tumor model and MC38 tumor wouldn’t grow in the other strains of mice. MC38 originated from B6 background mice and can only be given to B6 mice. 

90 Please clarify the timepoint of CFA injection (e.g. day x after first CPI treatment/tumor injection etc.)

Thank you for this comment. We have clarified the timepoint in the text. CFA was given on days 35 and 56 after initial ICI injection. 

143 Please specify if this is the case in all strains/with all treatments.

Thank you for the comment. We have altered the text to reflect the effect on the mice strains. CFA failed to induce immune infiltration in BALB/c strain.

147 Based on IHC, please specify the types of infiltrating mononuclear cells.

Thank you for this comment. The types of infiltrating mononuclear were CD4+ T cells and CD19+ B cells. Macrophages staining (F4/80) was added to the revised version. 

157 Please state if CFA was injected in both ‘treated’ and ‘untreated’ mice.

CFA was not injected in this experiment. As mentioned in the text, CTLA-4 and PD-1 were injected in combination.

163 This is an interesting finding that should be further discussed as it includes sites of most frequent irAE in humans and might – or might not – be due to effects mediated by other cells and mechanisms than T cells.

We agree with the reviewer. The infiltrations in these sites might or might not be directly related to T cells, and this is now further discussed. Thank you. 

175 Specify ‘immune infiltration’ e.g. of tumors or off-target organs.

This was fixed (line 180-181). Thank you.

183 This relates to figure 2F, not 2D

Thank you so much, this has been corrected.

212 Please specify if there is a segregation within the group concurrently treated with prednisolone!

Thank you for the comment. The text has been edited to mention this. There was segregation within the group treated with ICI and prednisolone, however, the tumor growth curve was a median between the non-responders and responders.

219 Less strong terminology than ‘no’ would be adequate as there are some limited baseline predictive markers such as levels of BP180 antibodies.

Thank you for the comment. The text has been edited and bullous pemphigoid antibodies should be considered and now are also appropriately cited in our text (Ali and Flatz. J Am Acad Dermatol 2020). 

220 This sentence is not easily understood.

This has been corrected.

232 This sentence is hard to understand.

The sentence has been corrected.

252 It should be stated whether this is found in treated tumors.

The text has been updated. 

312 In lieu of references 32/33, reports of pivotal trials or robust review articles might be preferable.

Thank you, references have been updated.

Table 2 Immune infiltrate to the colon is reported here as opposed to the absence of infiltrate reported in the results section and in Figure 1.

Thank you. This was a mistake that was fixed in the revised version. There was immune infiltrate to the colon in the treated mice. 

Figure 4. The title is misleading, as prednisolone is reducing both tumor control and T cell infiltration in other organs in this experiment.

Indeed, the title has been changed. Thank you.

---

## [Editor Report · Decision Letter 1]

15 Jan 2021

A novel mouse model for checkpoint inhibitor-induced adverse events

PONE-D-20-17843R1

Dear Dr. Mor,

We’re pleased to inform you that your manuscript has been judged scientifically suitable for publication and will be formally accepted for publication once it meets all outstanding technical requirements.

Kind regards,

Prof. Pierre Bobé

Academic Editor

PLOS ONE

---

## [Editor Report · Acceptance letter]

26 Jan 2021

PONE-D-20-17843R1 

A novel mouse model for checkpoint inhibitor-induced adverse events 

Dear Dr. Mor:

I'm pleased to inform you that your manuscript has been deemed suitable for publication in PLOS ONE. Congratulations! Your manuscript is now with our production department. 

Kind regards, 

on behalf of

Prof Pierre Bobé 

Academic Editor

PLOS ONE